# Inhibitor of CD147 Suppresses T Cell Activation and Recruitment in CVB3-Induced Acute Viral Myocarditis

**DOI:** 10.3390/v15051137

**Published:** 2023-05-10

**Authors:** Ruifang Wang, Kexin Zong, Juan Song, Qinqin Song, Dong Xia, Mi Liu, Haijun Du, Zhiqiang Xia, Hailan Yao, Jun Han

**Affiliations:** 1State Key Laboratory of Infectious Disease Prevention and Control, National Institute for Viral Disease Control and Prevention, Chinese Center for Disease Control and Prevention, 155 Changbai Rd., Beijing 102206, China; rui6051990@163.com (R.W.); gfyssyszkx@163.com (K.Z.); helen831020@126.com (J.S.); sanban1605@163.com (Q.S.); xdgoforit@126.com (D.X.); liumi@ivdc.chinacdc.cn (M.L.); duhj@ivdc.chinacdc.cn (H.D.); xiazhiqiang929@aliyun.com (Z.X.); 2Department of Biochemistry & Immunology, Capital Institute of Pediatrics, YaBao Rd., Beijing 100020, China

**Keywords:** CD147, anti-inflammation, T cell activation, Th cell subsets, viral myocarditis

## Abstract

Viral myocarditis (VMC) is a common disease characterized by cardiac inflammation. AC-73, an inhibitor of CD147, disrupts the dimerization of CD147, which participates in the regulation of inflammation. To explore whether AC-73 could alleviate cardiac inflammation induced by CVB3, mice were injected intraperitoneally with AC-73 on the fourth day post-infection (dpi) and sacrificed on the seventh dpi. Pathological changes in the myocardium, T cell activation or differentiation, and expression of cytokines were analyzed using H&E staining, flow cytometry, fluorescence staining and multiplex immunoassay. The results showed that AC-73 alleviated cardiac pathological injury and downregulated the percentage of CD45^+^CD3^+^ T cells in the CVB3-infected mice. The administration of AC-73 reduced the percentage of activated CD4^+^ and CD8^+^ T cells (CD69^+^ and/or CD38^+^) in the spleen, while the percentage of CD4^+^ T cell subsets in the spleen was not changed in the CVB3-infected mice. In addition, the infiltration of activated T cells (CD69^+^) and macrophages (F4/80^+^) in the myocardium also decreased after the AC-73 treatment. The results also showed that AC-73 inhibited the release of many cytokines and chemokines in the plasma of the CVB3-infected mice. In conclusion, AC-73 mitigated CVB3-induced myocarditis by inhibiting the activation of T cells and the recruitment of immune cells to the heart. Thus, CD147 may be a therapeutic target for virus-induced cardiac inflammation.

## 1. Introduction

Myocarditis is a common inflammatory cardiac disorder caused by infection, idiopathic disorders or autoimmune diseases. Myocarditis is also the main cause of dilated cardiomyopathy (DCM) and probably results in heart failure and sudden cardiovascular death, especially in adolescents. It is considered one of the most challenging clinical problems due to the lack of effective diagnosis and treatment [1]. Myocarditis can be caused by viral or bacterial infection, rheumatic carditis, sarcoidosis, overexpression of desmocollin-2 induced by a genetic disease, and toxic or hypersensitive drug reactions [2,3,4]. A viral infection is the most common cause of myocarditis, such as an infection by enterovirus, EB virus or human herpesvirus 6 [5,6]. Coxsackievirus B3 (CVB3), a cardiotropic virus, is one of the most important viruses causing viral myocarditis (VMC) [2,7].

The pathological progress of VMC can be divided into three stages at the cellular and tissue levels: the acute stage characterized by myocardial damage resulting from viral entry and replication, the subacute stage characterized by inflammatory cell infiltration, and the chronic stage characterized by cardiac remodeling [8]. In terms of immune responses, it is well known that CVB3 acts as an initiator of myocarditis and induces autoimmunity responses. In the acute stage, cellular immunity is triggered, which is manifested by the infiltration of natural killer cells and macrophages in the myocardium, thus aggravating myocardial injury. Subsequently, the infiltration of T cells, including CD4^+^ T and CD8^+^ T cells, induces inflammation or mediates tissue damage [9,10]. Previous studies have also shown that CVB3 infection induces the activation and differentiation of T cells, as well as the secretion of chemokines, including MCP-1, IP-10, MIP-1 and MIP-2, and inflammatory cytokines [11,12,13].

Many studies have shown that CD147, also called EMMPRIN, participates in T cell activation and recruitment [14,15,16]. Additionally, some researchers have indicated that CD147 mAb can reduce inflammation in multiple sclerosis, acute asthmatic inflammation and rheumatoid arthritis by regulating T cell reactivity [17,18,19]. One study showed that the expression of CD147 in the heart upregulates four days post CVB3 infection [14]. These results suggest that CD147 may participate in the pathogenesis of CVB3-induced VMC by regulating immune cell infiltration. The interaction between CD147 on inflammatory cells and the extracellular ligand, Cyclophilin A (CyPA), further promotes the infiltration of inflammatory cells [15]. Previous studies have revealed that CyPA is upregulated in myocardium after CVB3 infection [14]. Some reports also show that CyPA plays an essential role in virus proliferation in HIV, HCV and EV71 infections [20,21]. In addition, the dimerization of receptors, such as CD147, is known to be an essential mechanism of signaling initiation [22,23]. Therefore, to understand the effect of CD147 on the immune damage mechanism of CVB3-induced myocarditis, AC-73, an inhibitor that interferes with the formation of CD147 dimerization [23], was administered to a mouse myocarditis model induced by CVB3 infection. AC-73 disrupts the dimerization of CD147, which may affect the signal of CD147-CyPA. Subsequently, pathological changes, including inflammatory cell infiltration, activation or differentiation of T cells, and expression of cytokines were observed and detected using H&E staining, flow cytometry, multiplex immunoassay and q-PCR in this study.

## 2. Materials and Methods

### 2.1. Animals

Male Balb/c mice (5–6 weeks old) weighing 18–20 g were provided by the Beijing Vital River Laboratory Animal Technology. They were housed under a pathogen-free facility and 12 h light/dark cycles with free access to food and water ad libitum. This work was licensed by the Laboratory Animal Ethics Committee of the National Institute for Viral Disease Control and Prevention, Chinese Center for Disease Control and Disease (China CDC). All experimental procedures were conducted in accordance with the National Institutes of Health (NIH) guidelines (*Guide for the Care and Use of Laboratory Animals, 8th Edition*, 2011).

### 2.2. Virus and CVB3 Infection

HeLa cells were maintained in Dulbecco’s Modified Eagle medium (DMEM) supplemented with 10% fetal bovine serum (FBS), 100 units/mL of penicillin, and 100 μg/mL of streptomycin. The cell line was purchased from the American Type Culture Collection (ATCC). CVB3 (Woodruff variant) was propagated by HeLa cells with 2% FBS as described previously [24]. Additionally, the stock virus (5.13 × 10^7^ TCID50/mL) was stored at −80 °C. Susceptible mice were inoculated intraperitoneally with CVB3 at 5.13 × 10^3^ TCID50/100 μL (the dose of virus was determined based on the LD50 results).

### 2.3. Treatment with AC-73

AC-73 was injected intraperitoneally (i.p.) on the 4th day after CVB3 infection. After the mice were sacrificed on the 7th day post-infection (dpi), the heart, spleen and blood were taken for analysis. The AC-73-treated group was divided into two subgroups based on the different doses (25 mg/kg and 50 mg/kg) of AC-73 administered [25].

### 2.4. Viral Genome Load and CD147 mRNA Expression in the Heart

The total RNA in the hearts of 5 mice in each group was extracted from the homogenized hearts using Trizol reagent. CVB3 RNA and CD147 mRNA were detected based on real-time quantitative polymerase chain reaction (q-PCR) using the Light Cycler 480 real-time PCR instrument (Roche, Basel, Switzerland). The primer sequence was used as previously described [26,27]. Then, the relative expression of CVB3 RNA and CD147 mRNA were normalized to the level of GAPDH mRNA using the 2^−∆∆CT^ method.

### 2.5. Histopathological Analysis

The hearts of 5 mice in each group were obtained and fixed in 4% paraformaldehyde for 24 h at room temperature. After being paraffin-embedded and sectioned, the sections were stained with hematoxylin and eosin (H&E) and then scanned using Pannoramic DESK (3D HISTECH, Budapest, Hungary). The infiltrated inflammation cells were visualized using the Caseviewer CV2.3 software (3DHISTECH, HUN). The severity of myocardial injury was judged based on the pathological score as described previously [13,28,29,30]. The myocarditis scoring ranging from 0 to 4 is as follows: 0 indicates no inflammatory infiltrates; 1 indicates small foci of inflammatory cells; 2 indicates larger foci > 100 inflammatory cells; 3 indicates ≤5% of the cross section involved; and 4 indicates >5% of the cross section involved.

### 2.6. Fluorescence Staining

After antigen retrieval, the sections were permeated with 0.3% Triton X-100. Then, the sections were stained with antibodies, including rabbit anti-mouse CD147 (1:50, Abcam, Cambridge, UK), rabbit anti-mouse F4/80 (1:500, Abcam), and rabbit anti-mouse CD69 (1:200, Abcam) overnight at 4 °C. After rinsing, the sections were incubated with fluorescent-labeled secondary antibodies at 37 °C for 1 h. Then, the sections were stained with DAPI (1 μg/mL) at room temperature for 1 h. The fluorescence images of the targeting proteins were viewed using a confocal microscopy (LEICA TCS SP8, Wetzlar, Germany). The integral optical density (IOD) values of each field-specific fluorescence staining were collected.

### 2.7. Immunohistochemistry

After antigen retrieval, the sections were quenched in 3% H_2_O_2_ for 10 min and blocked with 5% BSA for 15 min. Then, the sections were incubated with rabbit anti-mouse CyPA (1:500, Abcam) overnight at 4 °C. Subsequently, the sections were incubated with HRP-conjugated goat anti-rabbit antibody at 37 °C for 1 h and visualized after incubation with 3,3′-diaminobenzidine tetrahydrochloride (DAB). Then, the sections were counterstained with hematoxylin. The immunohistochemical images were viewed using an Olympus BX51 fluorescence microscope (Olympus, Tokyo, Japan) and analyzed using the Image-Pro Plus 6.0 software (Media Cybernetics, Rockville, MD, USA). The value of the area density of the images was used for statistical analysis.

### 2.8. FACS Analysis for T Cell Subtype

The spleen of 5 mice in each group was homogenized and filtered using a 70 μm strainer to obtain single-cell suspension. Then, erythrocytes were lysed and spleen mononuclear cells were obtained. At the same time, the hearts (15 mice in each group, 5 mice/sample) were minced and digested with multiple enzymes complex (200 μg/mL of collagen Ⅱ and 50 μg/mL of hyaluronidase) at 37 °C with 200 rpm shaking for 1 h. Then, the obtained cells were filtered, and mononuclear cells were harvested at 40% and 70% of Percoll following the gradient centrifugation procedure. The mononuclear cells isolated from the spleens and hearts were stimulated with PMA and ionomycin for 6 h and then stained with Brilliant Violet 605™ anti-mouse CD45 (1:100), PE/Dazzle™ 594 anti-mouse CD3ε (1:100), PerCP/Cy5.5 anti-mouse CD4 (1:500), APC/Cyanine7 anti-mouse CD8b (1:50), Brilliant Violet 421™ anti-mouse IL-17A (1:50), PerCP/Cyanine5.5 anti-mouse IL-4 (1:25), Alexa Fluor^®^ 700 anti-mouse IFN-γ (1:20), PE/Cyanine5 anti-mouse CD25 (1:250), Alexa Fluor^®^ 647 anti-mouse FOXP3 (1:50), Brilliant Violet 711™ anti-mouse CD69 (1:50), and PE/Cyanine7 PE/Cyanine7 CD38 (1:500). The samples were acquired using a BD FACSLSR Fortessa flow cytometer (BD Biosciences, Franklin Lakes, NJ, USA), and data analysis was performed using the FlowJo softwareV10 (BD Biosciences). The representative gating strategy for immune cells is presented in Appendix A.

### 2.9. Luminex Multiplex Immunoassay for Cytokine and Chemokine Profiling

The blood of 5 mice in each group was collected into a 0.5 mL anticoagulant tube from the eye socket. Within 30 min, the blood samples were centrifuged for 10 min at 1000× *g* to obtain plasma. Then, the obtained plasma was stored at −80 °C. Afterward, the frozen plasma was thawed on ice and centrifuged at 10,000× *g* for 10 min, and the supernatant was taken for detection. Overall, 15 cytokines and chemokines were detected following the manufacturer’s instructions using a ProcartaPlex Mouse Cytokine/Chemokine Panel 1A 36 Plex (EPX360-26092-901, Invitrogen, Waltham, MA, USA) with the Luminex^TM^ 200^TM^ instrument. The data were normalized based on the z-score. Then, the data were visualized and hierarchical cluster analysis was performed to obtain a heatmap using Rstudio.

### 2.10. Cytokine mRNA Expression

The RNA obtained from the homogenized hearts was transcribed into cDNA using a PrimeScript™ RT reagent Kit (Takara, Tokyo, Japan). Then, q-PCR analyses were performed using a TB Green Premix Ex Taq II Kit (Takara, Japan). The primer sequences are presented in Appendix A [31,32,33]. The results were expressed using the 2^−ΔΔCt^ method.

### 2.11. Statistical Analysis

All data are presented as mean ± standard deviation (SD). The statistical software employed was Graphpad 9.0 (GraphPad software, Boston, MA, USA). The data were subjected to normality tests and tests of homogeneity of variances based on the Shapiro–Wilk test and the Brown–Forsythe test. Then, according to the statistical characteristics, the data were analyzed using the Kruskal–Wallis test or one-way analysis of variance (ANOVA). Values of *p* < 0.05 was defined to indicate statistically significant results. * *p* < 0.05, ** *p* < 0.01, *** *p* < 0.001 and **** *p* < 0.0001 was used to set statistical significance.

## 3. Results

### 3.1. AC-73 Did Not Affect the Expression of CD147 Induced by CVB3 Infection

To understand whether CVB3 infection induced myocardial expression of CD147, each Balb/c mouse was inoculated intraperitoneally (i.p.) with CVB3 at 5.13 × 10^3^ TCID50/100 μL. Then, myocardial expression of CD147 mRNA was detected on the 7 dpi. The results showed that the expression of CD147 mRNA increased significantly in the CVB3-infected group (7.30 ± 1.12) compared to the control group (1.04 ± 0.26) (*p* < 0.01). To see whether AC-73 affected the expression of CD147 induced by CVB3 infection, the Balb/c mice were administrated (i.p.) 25 mg/kg or 50 mg/kg of AC-73 on the fourth dpi according to a previous research study [25]. After the administration of two doses of AC-73, the relative expression of CD147 mRNA decreased to 6.38 ± 1.42 and 5.42 ± 0.26, respectively, for the mice administered 25 mg/kg and 50 mg/kg of AC-73. Two doses of AC-73 caused a slight decrease in the expression of CD147 mRNA in the myocardium of the CVB3-infected mice, although there was no statistical difference when compared with the CVB3-infected mice (*p* > 0.05) (Figure 1A).

Meanwhile, the effect of AC-73 on CD147 or CyPA expression induced by CVB3 infection was evaluated using immunofluorescence and immunohistochemistry. The immunofluorescence results showed that CVB3 infection induced an increase in the relative integral optical density (IOD) of CD147 (1.54 ± 0.22) compared to the control (1.00 ± 0.09) (*p* < 0.001), whereas treatment with AC-73 did not inhibit the CD147 expression induced by CVB3 infection (Figure 2A,C). Additionally, the immunohistochemical results showed that CVB3 infection also mediated an increase in the average density of CyPA (55.12 ± 6.83) compared to the control (8.36 ± 1.90) (*p* < 0.001), while treatment with AC-73 reduced the average density of CyPA due to CVB3 infection, especially in the 50 mg/kg AC-73 group (13.79 ± 3.40) (*p* < 0.001) (Figure 2).

Previous research studies have reported that the CD147 ligand, CyPA, prohibits the replication of many viruses, such as HIV and HCMV [34,35]. To know whether AC-73 inhibited CVB3 virus replication in the mice’s myocardium, the viral load was detected using q-PCR after the administration of 50 mg/kg of AC-73 or 25 mg/kg of AC-73. Compared to the CVB3 model group (1.02 ± 0.24), the viral load in the myocardium of the 25 mg/kg AC-73 (1.36 ± 0.40) or 50 mg/kg AC-73 (1.22 ± 0.59) group did not show statistically significant difference (Figure 1B).

### 3.2. AC-73 Alleviated Pathological Injury in the Heart

The above results showed that the body weight of the mice decreased with a prolongation of infection time after CVB3 infection. However, both 50 mg/kg of AC-73 and 25 mg/kg of AC-73 improved the weight loss due to CVB3 infection, although there was no statistical difference between the 25 mg/kg AC-73 group and the CVB3-infected group (*p* > 0.05). The CVB3-infected mice lost 26.78 ± 2.80% of their body weight, and the AC-73-treated mice treated with 50 mg/kg lost 18.14 ± 3.64% of their body weight (Figure 1C).

Then, H&E staining was used to determine whether AC-73 affected myocardial pathological changes in the mice on the seventh dpi. The results showed that inflammatory infiltration and myocardial necrosis could be observed in the myocardium of the CVB3-infected mice. The AC-73 treatment significantly reduced the infiltration of inflammatory cells, and the mice treated with AC-73 displayed less myocardial necrosis than the CVB3-infected mice. The pathological score of the myocardium in the CVB3-infected mice increased on the seventh dpi. However, the administration of 50 mg/kg of AC-73 decreased the score when compared with the CVB3-infected mice (*p* < 0.01). Additionally, the myocardial pathological injuries caused by CVB3 infection improved with an increase in AC-73 dosage (Figure 3). Myocardium damages were not found in the mice treated with 50 mg/kg AC-73 (Figure 3B). The results demonstrated that AC-73 alleviated the myocardial pathological injuries of the CVB3-infected mice.

### 3.3. AC-73 Reduced the Proportion of Activating T Cells Induced by CVB3 Infection

Previous studies report that CD147 participates in T cell activation and recruitment [36]. To know whether CD147 was involved in the T cell activation and recruitment of the CVB3-infected mice, the function and proportion of T cells in the spleen and myocardium of the CVB3-infected mice were analyzed after being exposed to AC-73. Firstly, the results showed that CVB3 infection induced an increase in total T cells (CD45^+^CD3^+^) from 44.57 ± 2.15% (control group) to 60.43 ± 1.91% (CVB3 model group) (*p* < 0.001). However, after the administration of 50 mg/kg of AC-73 or 25 mg/kg of AC-73, the percentage of total T cells recovered from 60.43 ± 1.91% (CVB3 model group) to 52.53 ± 2.86% or 52.37 ± 1.76%, respectively in the spleen (*p* < 0.05). The ratio of CD4^+^/CD8^+^ T cells in the spleen showed no significant difference among the four groups (Figure 4A,C). To understand the effect of AC-73 on T cell response, activated CD4^+^ and CD8^+^T cells in the mouse spleen were detected using flow cytometry on the seventh dpi. The results showed that CVB3 infection induced an increase in activating T cells (CD38^+^ and/or CD69^+^ T cells) among CD4^+^ or CD8^+^ T cells in the spleen compared to the control group (*p* < 0.05). CD69^+^ CD4^+^T cells (early activated T cells) and CD38^+^ CD4^+^T cells induced by CVB3 infection accounted for 7.52 ± 0.96% and 11.03 ± 0.58% of CD4^+^ T cells, respectively. Similarly, both 50 mg/kg of AC-73 and 25 mg/kg of AC-73 also inhibited the increase in activating CD4^+^ T cells and CD8^+^ T cells due to CVB3 infection. After the mice were treated with 50 mg/kg of AC-73 or 25 mg/kg of AC-73, the percentage of early activated CD69^+^ CD4^+^ T cells decreased to 4.38 ± 0.24% (*p* < 0.01) or 6.94 ± 0.51% (*p* > 0.05), respectively. Additionally, CD38^+^ CD4^+^ T cells decreased to 6.76 ± 0.37% or 8.42 ± 0.43%, respectively (*p* < 0.05). CD69^+^ CD8^+^ T cells (early activated CD8^+^T cells) and CD38^+^ CD8^+^T cells induced by CVB3 infection accounted for 3.87 ± 0.67% and 4.88 ± 0.45% of CD8^+^ T cells in this study. After being exposed to 50 mg/kg of AC-73, the percentage of CD38^+^ CD8^+^ T cells and CD69^+^ CD8^+^ T cells decreased to 3.44 ± 0.51% and 2.77 ± 0.50% among the CD8^+^ T cells (*p* < 0.05), respectively (Figure 5). In summary, AC-73 decreased the proportion of activated CD4^+^/CD8^+^ T cells among the total CD4^+^/CD8^+^ T cells. The above results showed that CVB3 infection induced an increase in both T cell response and inflammatory reaction on the seventh dpi, while AC-73 inhibited the T cell response induced by CVB3 infection. However, the AC-73 treatment did not change the proportion of CD4^+^ T cell subsets in the spleen, such as Th1, Th2, Th17 and Treg in the CVB3-infected mice (Figure 6). In addition, the analysis of myocardial tissue resident T cells found that AC-73 treatment decreased the proportion of T cells significantly from 75.63 ± 1.43% (CVB3-infected group) to 38.07 ± 6.11% (25 mg/kg of AC-73) or 34.00 ± 6.52% (50 mg/kg of AC-73) in the heart (*p* < 0.0001). The AC-73 treatment did not impact on the proportion of CD4^+^ T cells and CD8^+^ cells of total T cells (Figure 4B,D).

To see the effect of AC-73 on the expression of CD69 in myocardium, the sections of the hearts harvested on the seventh dpi were stained with rabbit anti-CD69 antibodies using immunofluorescence. The results showed that CVB3 infection induced an increase in the relative IOD of CD69 (4.18 ± 0.73) compared to the control (*p* < 0.001). However, treatment with 25 mg/kg of AC-73 (1.94 ± 0.48) decreased the expression of CD69 in the myocardium induced by CVB3 infection, although it was higher than that of the control (*p* < 0.05). The expression of CD69 in the 50 mg/kg AC-73 group (1.11 ± 0.18) was lower compared to the 25 mg/kg AC-73 group (*p* < 0.05) (Figure 7) The AC-73 treatment reduced the expression of CD69, a T cell activation marker, in the infected mice.

### 3.4. AC-73 Inhibited the Infiltration of Macrophages in CVB3-Infected Mice

To understand whether AC-73 treatment can affect other immune cells in CVB3 infected mice, the macrophages were detected in myocardium using immunofluorescence. The heart sections of 7 dpi were stained by rabbit anti-F4/80 antibodies. The results showed that CVB3 infection caused the increase in the macrophages in myocardium (5.41 ± 0.44) compare to the control (*p* < 0.001). However, treatment with AC-73 decreased the infiltration of macrophages in myocardium induced by CVB3 infection. The macrophages in 50 mg/kg AC-73 group (1.31 ± 0.21) was less than that in 25 mg/kg AC-73 group (2.20 ± 0.37) (*p* < 0.05) (Figure 8).

### 3.5. AC-73 Inhibited the Expression of Cytokines and Chemokines in CVB3-Infected Mice

To understand whether the AC-73 treatment could affect the expression of cytokines in the CVB3-infected mice, the expression of 15 cytokines were detected in the plasma. The results showed that many chemokines and cytokines were upregulated in the CVB3-infected group. The expression of the chemokines, such as MCP-1, MIP-1, MIP-2 and RANTES, and cytokines, such as IFN-γ, TNF-α, IL-2, IL-1β, IL-4, IL-10 and IL-17, increased obviously on the seventh dpi. However, compared to infected group, the expression of all of these cytokines decreased in the AC-73 treated group (Figure 9A).

Then, the expressions of myocardial IFN-γ, IL-10 and IL-17 mRNA were detected using q-PCR. The results showed that the mRNA expression of IFN-γ, IL-17 and IL-10 in the myocardium of the CVB3-infected group was 4.40-fold, 8.74-fold and 8.30-fold higher than that of the control group, respectively (*p* < 0.05). However, both 50 mg/kg of AC-73 and 25 mg/kg of AC-73 treatments inhibited the expression of IFN-γ, IL-17 and IL-10 mRNA in the myocardium of the CVB3-infected mice (Figure 9B).

## 4. Discussion

After CVB3 infects cardiomyocytes, viral proteases (2A, 3C) destroy the integrity of myocardial skeleton and sarcolemma, inhibit the synthesis of cardiomyocytes, and cause apoptosis, degeneration and necrosis of cardiomyocytes [37,38]. These are the mechanisms by which viruses directly cause myocardial pathological damage. Subsequently, immune impairment occurs. Indeed, the viruses themselves may not be the direct agents eliciting inflammation of the heart, while immune reaction is associated with inflammation. The products of destroyed cells induced by immune response, including proinflammatory cytokines and proteolytic enzymes, could also be toxic to the host cells. Myocarditis is regulated by “the protein-homeostasis-system (PHS)”, with the immune system being considered one part of the PHS of the host [39,40]. At the acute stage of VMC, cellular immunity is triggered to enhance the host’s autoimmunity and defense function. The cellular immunity of acute VMC is divided into two phases. In phase I, natural killer cells and macrophages infiltrate the myocardium, which is an early anti-virus barrier, and can also aggravate myocardial damage. Phase II involves mainly T cell infiltration, which is characterized by the inflammatory reaction caused by cytokines released by CD4^+^ T cells and the toxic effect mediated by CD8^+^ T cells [41,42]. Previous studies revealed that the peak time point of acute myocarditis occurred on the seventh day after infection in a CVB3 mouse model [43], and CD147 was upregulated in the myocardium four days after infection [14]. Therefore, AC-73 was intraperitoneally injected into the mice on the fourth day after infection. Additionally, the effects of AC-73 on myocardial inflammation on the seventh dpi were explored. Indeed, our results showed that CVB3 infection induced an increase in total T cells (CD45^+^CD3^+^) and activated T cells compared to the control group. In this study, our results revealed that AC-73, an inhibitor of CD147, improved the myocardial pathological damage and weight loss of the CVB3-infected mice. AC-73 could also prevent the release of inflammatory factors and the infiltration of T cells induced by CVB3. In addition, macrophages infiltrated in the myocardium also reduced after the AC-73 treatment. AC-73 inhibited T cell activation and T cell recruitment to the heart instead of affecting the differentiation of splenic CD4^+^ T cell subsets in our CVB3-induced myocarditis model.

As an upstream regulator of cardiac inflammation, CD147 reflects an increase in inflammatory activity. The expression of CD147 is enhanced in virus-positive myocardium, which is supposed to reflect increased inflammatory activity [44,45]. A study showed that CVB3 infection induces upregulation of CD147 in mouse cardiomyocytes [14]. Our results also illustrated that the expression of CD147 and CyPA was upregulated in myocardium after CVB3 infection.

Except for myocytes, CD147 is highly expressed on activated T cells or macrophages. CD147 plays an important role in the occurrence and development of inflammation, which has been proven by the construction of gene knockout mice using embryonic stem cells (ES) and homologous recombination technology. The knockout of CD147 showed abnormal lymphocyte response in the development of systemic lupus erythematosus and psoriasis [46,47,48]. In addition, CyPA was secreted upon inflammatory stimulation or released during cell death. Extracellular CyPA has chemotactic activity on neutrophils, eosinophils and T lymphocytes, while CD147 can act as a receptor of CyPA to mediate its chemotactic activity [49]. CyPA-CD147 interaction plays an important role in the initiation and progression of inflammatory response by inducing leukocytes in inflammatory tissues [15]. In vivo, anti-CD147 antibody can significantly reduce the inflammatory response in stroke-associated pneumonia [50]. Additionally, in allergic asthma and rheumatoid arthritis mouse models, CD147 antibody reduces the inflammatory reaction by more than 50% and 75%, respectively [18,51]. In this study, we found the inhibitor AC-73, which interferes with the formation of CD147 dimerization, also significantly inhibited the inflammatory infiltration of total T cells in the heart and activation of T cells in the CVB3-induced myocarditis model. At present, most of the studies showed that AC-73 affects the function of CD147 by disrupting signaling pathways, such as the MAPK/STAT3 pathway [25]. The dimerization of receptors is known to be an essential mechanism of signaling initiation. CD147 requires dimerization for the initiation of its signaling function [22]. AC-73 disrupts the dimerization of CD147, thereby affecting the initiation of a signaling function. Our results showed that the treatment with AC-73 also reduced the expression of CyPA. The function of CD147 in myocytes was disrupted by AC-73, which might affect the CD147-CyPA axis. Additionally, the inhibition of CD147, which is expressed on T cells or macrophages, might further affect the interaction of CD147 and extracellular CyPA.

We further explored the protective mechanism of AC-73 in VMC and found that the percentage of splenic T cells was decreased in the AC-73-treated group. However, compared to the CVB3 model group, the percentage of CD4^+^ or CD8^+^ among CD3^+^ T cells in the spleen showed no significant difference, which illustrated that the suppressive function for CD4 and CD8 was equal. Additionally, the administration of AC-73 also inhibited the activation of T cells in the spleen (CD69^+^CD4^+^/CD69^+^CD8^+^/CD38^+^CD4^+^/CD38^+^CD8^+^) induced by CVB3 infection. Considering that the increased expression and the dimerization of CD147 might enhance the CD147-CyPA interaction [16], the activation of T cells was suppressed by AC-73, which might disrupt the CD147-CyPA axis. Previous studies showed that the differentiation of CD4^+^ T cells plays a major role in the progress of VMC, including Th1 and Th17 in the spleen which may promote acute VMC [42,52,53]. However, the percentage of splenic CD4^+^ T cell subsets, including Th1/Th2/Th17/Treg, in the spleen showed no statistical difference between the model group and the AC-73-treated groups in this study. Therefore, AC-73 may affect T cell activation rather than the differentiation of CD4^+^ T cells.

The levels of T cells, natural killer cells and inflammatory cytokines in VMC patients indicate whether there are immune system disorders in VMC patients. To evaluate the immune response in CVB3-induced VMC, the expression of 15 cytokines in the plasma and the mRNA expression of 3 inflammatory cytokines, including IFN-γ, IL-10 and IL-17, in the heart were detected after the administration of AC-73. The results indicated that AC-73 significantly inhibited the upregulation of cytokines, including IL-1β, IFN-γ, TNF-α, IL-17, IL-4 and IL-10. AC-73 also suppressed the expression of chemokines, such as MCP-1, MIP-1, MIP-2 and RANTES, induced by CVB3 infection. AC-73 not only inhibited the expression of the above cytokines in the plasma of the CVB3-infected mice, but it also inhibited the mRNA expression of IFN-γ and IL-17 in the myocardium, which were considered to have a role in promoting the progress of viral myocarditis in previous studies [54]. Surprisingly, CVB3 infection promoted the expression of the anti-inflammatory factor IL-10 in the myocardium. We will further study the causes of IL-10 reduction. These results all demonstrated that AC-73 reduced myocardial inflammation in CVB3-induced VMC, although the expression of CD147 mRNA showed no difference between the CVB3-infected group and the AC-73-treated group, which were similar to previous research results on chronic colitis [55].

Previous studies found that the CD147 ligand, CyPA, is a host factor that benefits the proliferation of many viruses. CyPA promotes the uncoating process of HIV-1 by integrating with the virion and then binding to CD147. Additionally, the interaction of CyPA-CD147 may promote HIV-1 replication by regulating the cytoplasmic domain of CD147 protein which is signaling independent [56]. Additionally, CyPA interacts with H-I loop of capsid protein VP1, which affects the uncoating process during the entry step of EV71 [20]. We also found that the administration of AC-73 had no statistical effect on the viral load in the heart compared with the CVB3-infected group. The CyPA-CD147 interaction plays an important role in virus replication and inflammatory infiltration. The inhibition of CD147 may affect the expression or function of CyPA, which can affect virus replication. However, our results indicated that the administration of AC-73 did not seem to inhibit the load of CVB3 in the heart. Therefore, the effect of AC-73 on the expression or function of CyPA and the effect of CyPA or CD147 on CVB3 replication also need more attention.

In conclusion, our results indicated that the CD147 inhibitor, AC-73, alleviates inflammatory injury by suppressing the activation of T cells and recruitment of T cells or macrophages to the heart in a CVB3-induced myocarditis mouse model. The specific mechanism of AC-73 during the chronic stage of VMC needs to be further explored.

## Figures and Tables

**Figure 1 viruses-15-01137-f001:**
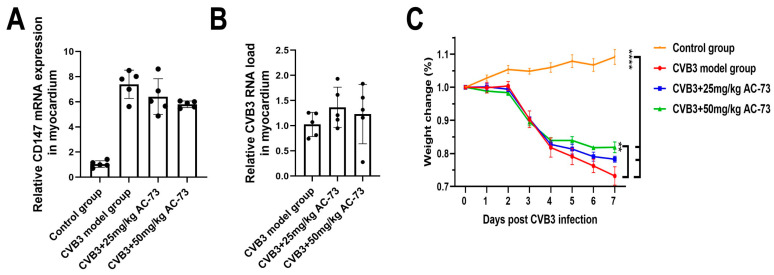
Effect of AC-73 on the expression of CD147 mRNA and virus replication in the myocardium of CVB3-infected mice and body weight. (**A**,**B**) The relative expression of CD147 mRNA and viral load in the myocardium on the 7th day post-infection was detected using q-PCR. The data were analyzed using one-way ANOVA, followed by Tukey’s multiple comparison post hoc tests (*n* = 5). (**C**) The weight change in the four groups. The data on weight were analyzed using one-way ANOVA, followed by Tukey’s multiple comparison post hoc tests (*n* = 5). ** *p* < 0.01, **** *p* < 0.0001.

**Figure 2 viruses-15-01137-f002:**
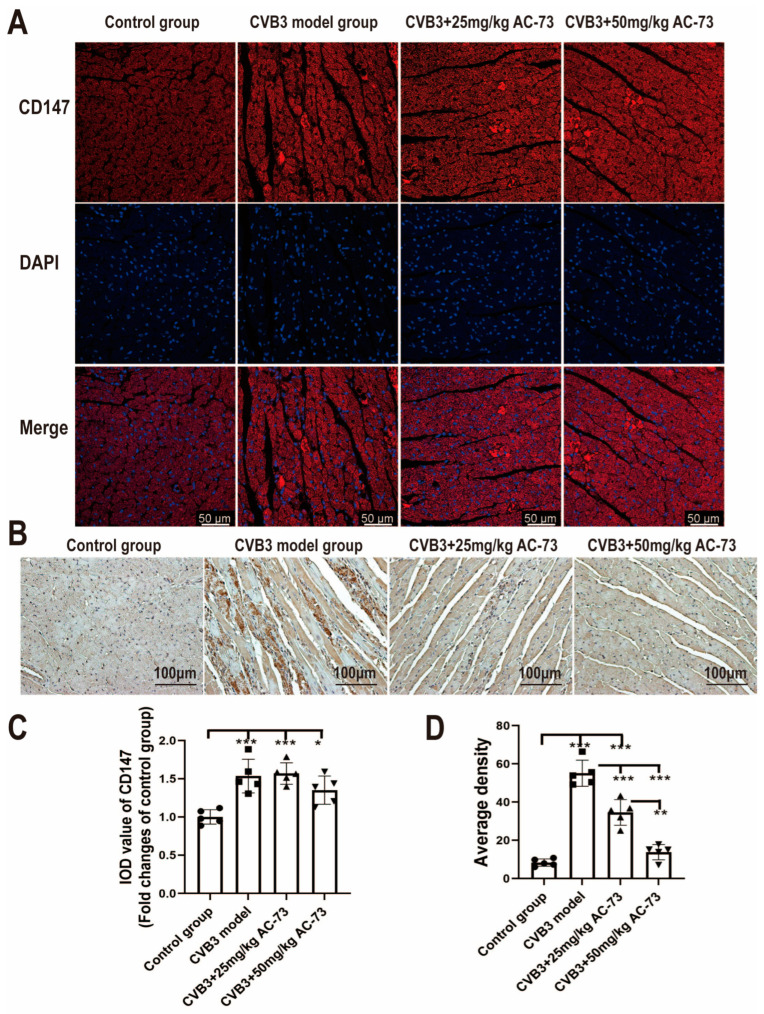
The effect of AC-73 on the expression of CD147 and CyPA in myocardium. The sections of the hearts on the 7th dpi were stained with rabbit anti-CD147 and rabbit anti-CyPA antibodies. (**A**) The immunofluorescence assays of CD147 in myocardium. Red indicates CD147 and blue indicates DAPI-stained cellular nuclei. Scale bar = 50 μm. (**B**) The immunohistochemical staining of CyPA in myocardium. Scale bar = 100 μm. (**C**,**D**) The histograms for the expression of CD147 and CyPA in myocardium. The data were analyzed using one-way ANOVA, followed by Tukey’s multiple comparison post hoc tests (*n* = 5). * *p* < 0.05, ** *p* < 0.01, *** *p* < 0.001.

**Figure 3 viruses-15-01137-f003:**
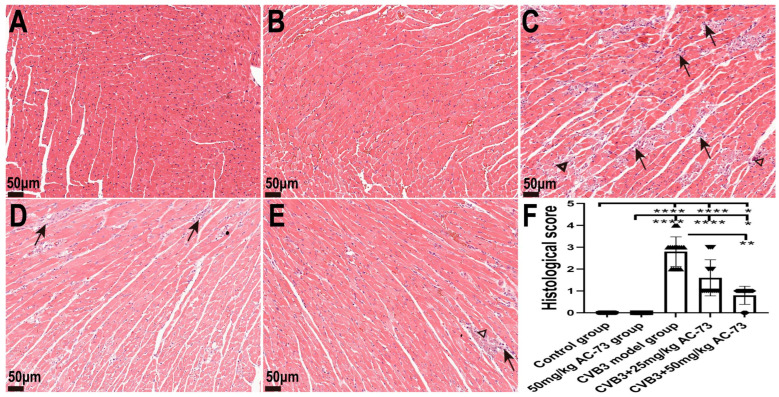
The histopathological change and the pathological score of the myocardium in the CVB3-infected mice after the administration of AC-73. Scale bar = 50 μm. The arrows represent inflammatory infiltration, and the triangles represent myocardial necrosis. All the mice were sacrificed on the 7th day post-infection and the hearts were taken for H&E staining: (**A**) control group; (**B**) 50 mg/kg AC-73 group; (**C**) CVB3-infected group; (**D**) CVB3-infected mice administered 25 mg/kg of AC-73; and (**E**) CVB3-infected mice administered 50 mg/kg of AC-73. (**F**) The pathological score of the myocardium (*n* = 5). Three visual fields were chosen randomly from each section, and the pathological score obtained using the three analyzed images from each heart tissue was statistically analyzed. The data were analyzed using the Kruskal–Wallis test, followed by Dunn’s multiple comparison test. * *p* < 0.05, ** *p* < 0.01, **** *p* < 0.0001.

**Figure 4 viruses-15-01137-f004:**
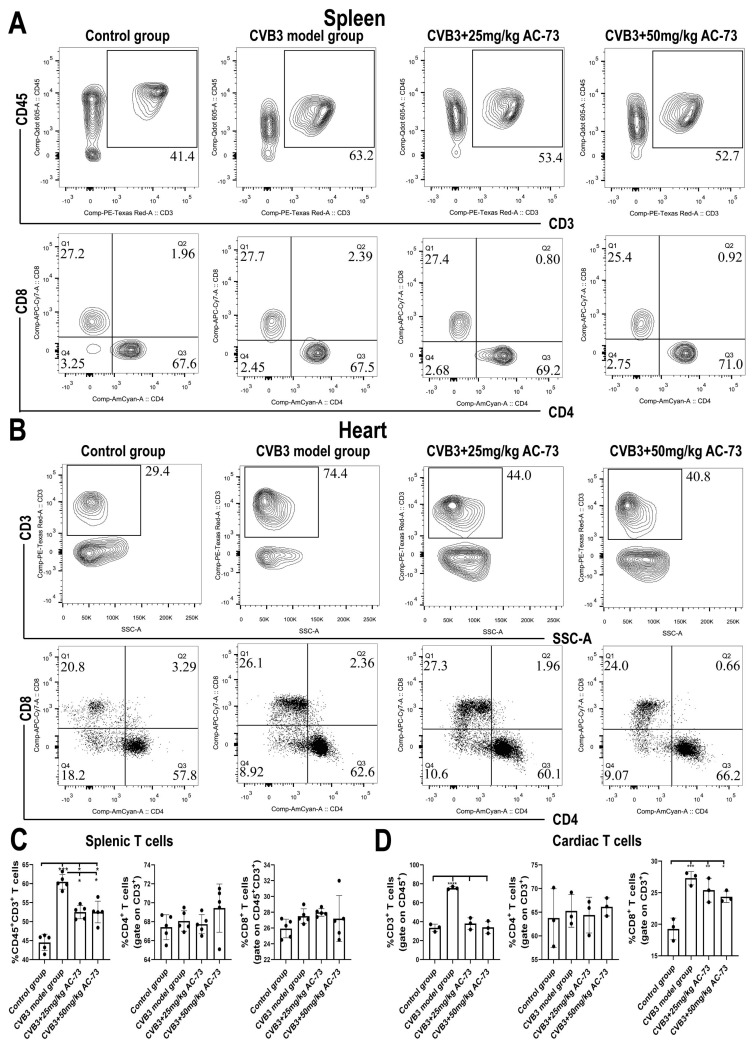
The proportion of total T cells, CD4^+^ cells or CD8^+^ T cells in the spleen and heart. After treatment with AC-73 on the 4th day after CVB3 infection, the mice were sacrificed on the 7th day post infection, and their spleen and heart were taken for flow cytometry. (**A**) The proportion of CD45^+^ CD3^+^, CD3^+^ CD4^+^ and CD3^+^ CD8^+^ T cells in the spleen. (**B**) The proportion of CD45^+^CD3^+^, CD3^+^CD4^+^ and CD3^+^CD8^+^ T cells in the heart. (**C**) The histograms for the percentage of T cells in the spleen (*n* = 5). (**D**) The histograms for the percentage of T cells in the heart (*n* = 3). The hearts from 15 mice in each group were obtained, and mononuclear cells of the hearts of every 5 mice were mixed into one sample (5 mice/sample); thus, we obtained 3 datasets in each group for statistical analysis. The data were analyzed using one-way ANOVA, followed by Tukey’s multiple comparison post hoc tests. * *p* < 0.05, ** *p* < 0.01, *** *p* < 0.001, **** *p* < 0.0001.

**Figure 5 viruses-15-01137-f005:**
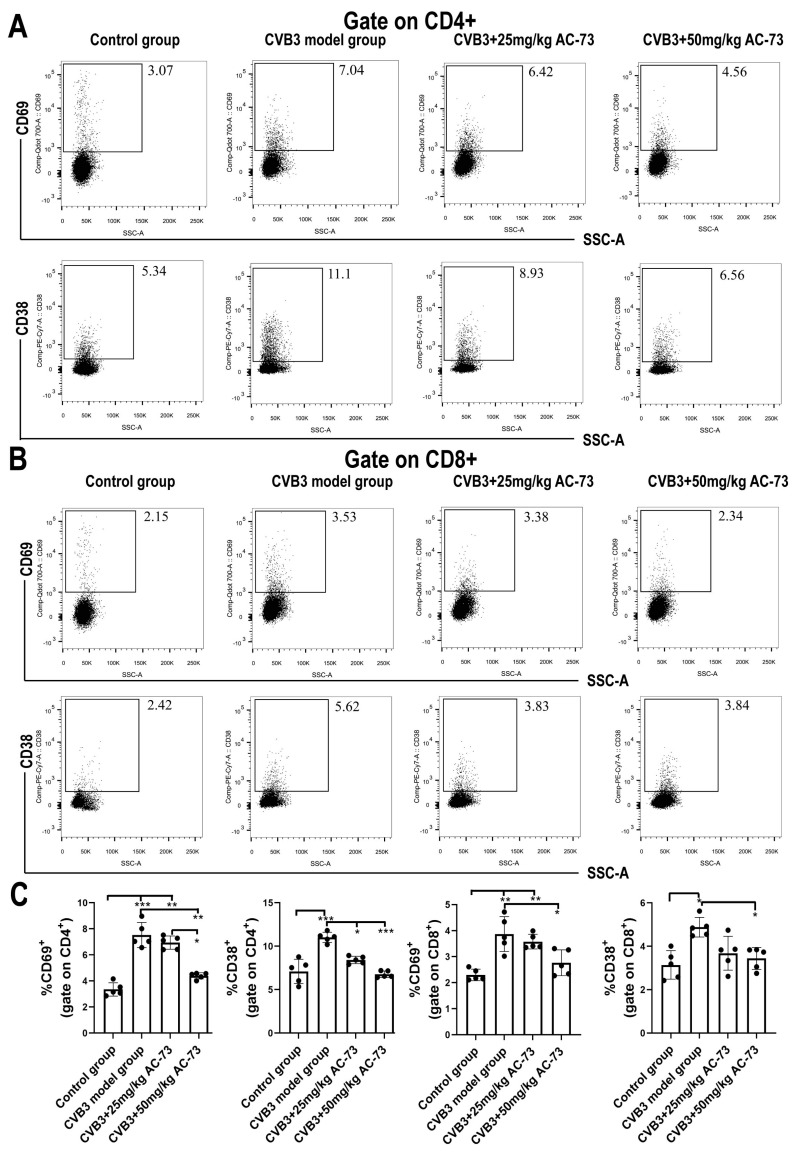
The proportion of activated T cells in the spleen. After treatment with AC-73 on the 4th day after CVB3 infection, the mice were sacrificed on the 7th day post-infection, and then spleen cells were taken for flow cytometry. (**A**) The proportion of CD69^+^/CD38^+^ CD4^+^ T cells in the spleen. (**B**) The proportion of CD69^+^/CD38^+^ CD8^+^ T cells in the spleen. (**C**) The histograms for the percentage of activated T cells in the spleen (*n* = 5). The data was analyzed using one-way ANOVA, followed by Tukey’s multiple comparison post hoc tests. * *p* < 0.05, ** *p* < 0.01, *** *p* < 0.001.

**Figure 6 viruses-15-01137-f006:**
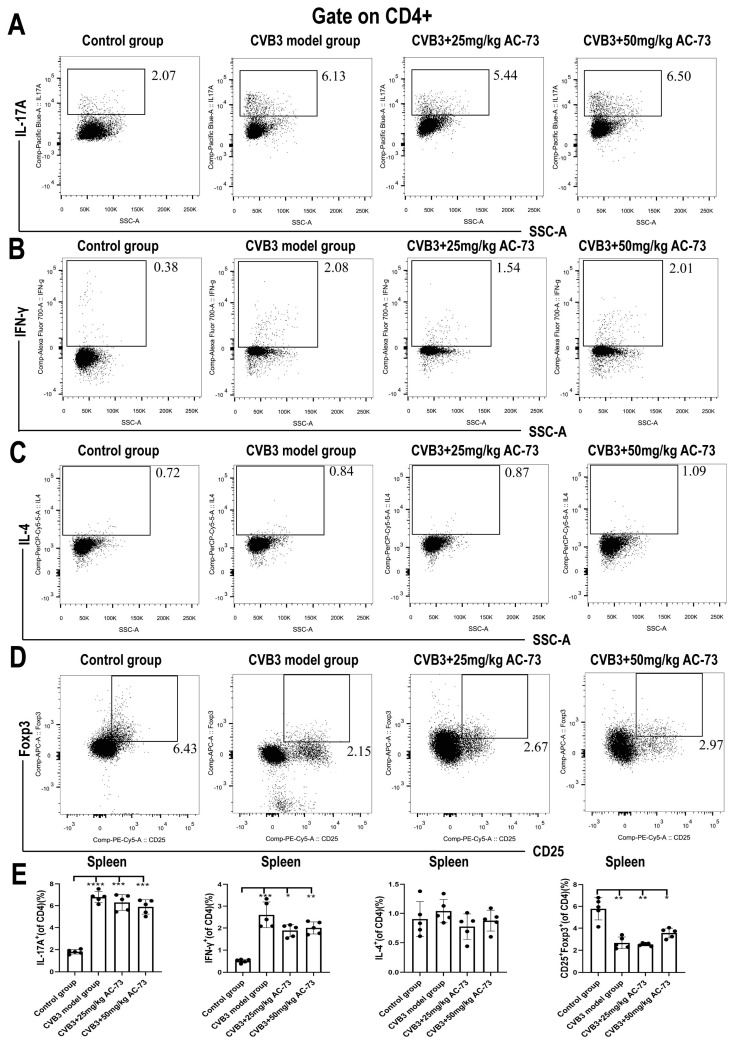
The proportion of Th17, Th1, Th2 and Treg cells in the spleen was detected using flow cytometry on the 7th day after CVB3 infection (*n* = 5): (**A**) the proportion of CD4^+^ IL17A^+^ T cells; (**B**) the proportion of CD4^+^ IFN-γ^+^ T cells; (**C**) the proportion of CD4^+^ IL-4^+^ T cells; and (**D**) the proportion of CD4^+^ CD25^+^ Foxp3^+^ T cells. (**E**) The histograms for the percent of Th17, Th1, Th2 and Treg cells in the spleen. The data were analyzed using one-way ANOVA, followed by Tukey’s multiple comparison post hoc tests. * *p* < 0.05, ** *p* < 0.01, *** *p* < 0.001, **** *p* < 0.0001.

**Figure 7 viruses-15-01137-f007:**
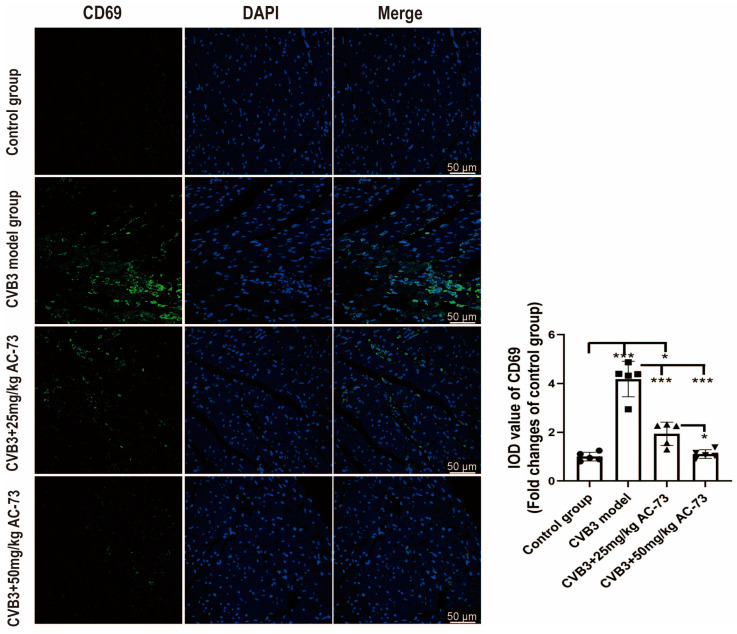
The effect of AC-73 on the expression of CD69 in myocardium. The sections of the hearts harvested on the 7th dpi were stained with rabbit anti-CD69 antibodies. Green indicates CD69, and blue shows DAPI-stained cellular nuclei. Scale bar = 50 μm. The data were analyzed using one-way ANOVA, followed by Tukey’s multiple comparison post hoc tests (*n* = 5). * *p* < 0.05, *** *p* < 0.001.

**Figure 8 viruses-15-01137-f008:**
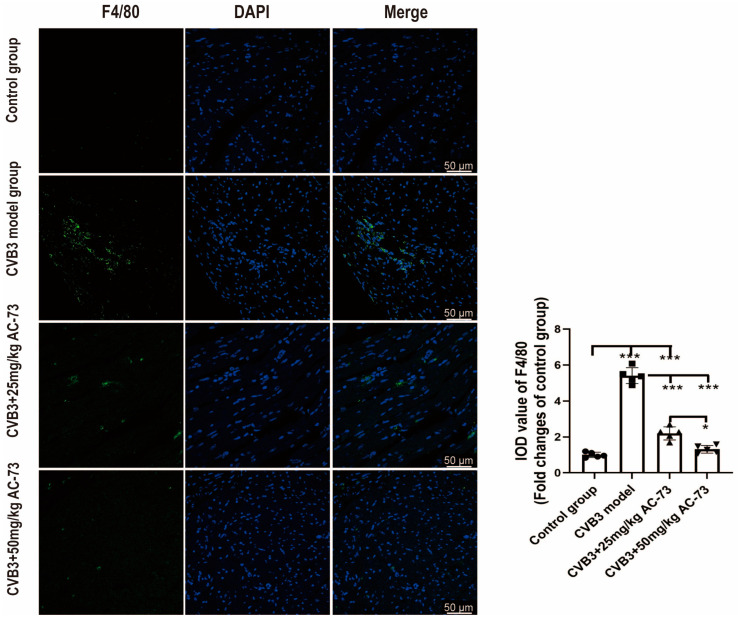
The effect of AC-73 on the macrophages in myocardium. The sections of hearts were stained with rabbit anti-F4/80 antibodies on 7 dpi. Representative images of macrophages stained with anti-F4/80 in myocardium. Green indicated F4/80 and blue showed DAPI-stained cellular nuclei. Scale bar = 50 μm. The data was analyzed by one-way ANOVA, followed by Tukey multiple comparison post hoc tests (*n* = 5). * *p* < 0.05, *** *p* < 0.001.

**Figure 9 viruses-15-01137-f009:**
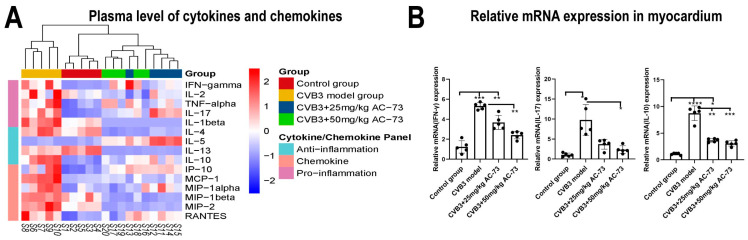
The effect of AC-73 on cytokine expression in the CVB3-infected mice. (**A**) Heat map of the expression of plasma cytokines and chemokines after the administration of AC-73 (*n* = 5). The 15 cytokines and chemokines were detected using Luminex multiplex immunoassay on the 7th day post-infection. In the heatmap, red and blue represent high and low expression, respectively. (**B**) The mRNA expression of IFN-γ, IL-10 and IL-17 in the heart was detected using q-PCR on the 7th day post-infection (*n* = 5). The data were analyzed using one-way ANOVA, followed by Tukey’s multiple comparison post hoc tests. * *p* < 0.05, ** *p* < 0.01, *** *p* < 0.001, **** *p* < 0.0001.

## Data Availability

Not applicable.

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
