# Peer review of "Inhibitor of CD147 Suppresses T Cell Activation and Recruitment in CVB3-Induced Acute Viral Myocarditis"

_viruses, 2023, doi:10.3390/v15051137_

Round 1

Reviewer 1 Report

Comments and Suggestions for Authors

General: 

This paper describes how the small molecule inhibitor AC-73, which inhibits CD147 dimerization, can greatly improve the pathological outcome of viral myocarditis. The authors focus on T cell infiltration and activation, and show that compared to mice infected with coxsackievirus B3 (CVB3), AC-73 treated mice T cell infiltration and activation, but not their subset frequencies, are reduced resulting in reduced cardiac inflammation and necrosis.

Major comments

1.       Introduction:  please explain why CD147 dimerization which is disrupted by AC-73, so important to its function, and specifically for its interaction with cyclophilin A (CypA) that is discussed at length in the discussion. 

2.       The experimental design is not well explained. Why was the endpoint of 7 days post infection chosen? Is the acute-chronic timeline well established in this model?

3.       CD147 expression was measured only by qPCR at the mRNA level, but its protein levels should also be explored. Moreover, the kinetics of CD147 expression should also be investigated in this model, to make sure that it peaks at 7 dpi.    

4.       Generally, the authors do not investigate the infiltration of macrophages at all. This is unexplained, and since CD147 is also expressed on macrophages and may be very important for in determining the cardiac microenvironment, this absence is critical.   

5.       There is a big difference between checking splenic T cells and determining their state of activation and changes in their subset frequencies, and showing the same data in cardiac infiltrating T cells. Although some effort was made in characterizing cardiac T cells, this was very partial. This reviewer appreciates that the task of isolating T cells from the heart is difficult, but at least it should be supported by some images of immunohistochemical or immunofluorescence staining of CD69 and CD38. 

6.       The determination of 15 cytokines and chemokines in the plasma is very interesting, but not sufficient. First, the pro-inflammatory cytokines and the chemokines should be explored in situ, that is in the heart tissue lysates.  Secondly, why did the authors specifically choose to investigate only IFNγ, IL-17 and IL-10? Moreover, since cytokines and chemokines are regulated at many different levels, the end-product of proteins should be explored, and not the mRNA.   

7.       The authors speculate a lot about the importance of the binding of CypA to CD147, and hypothesize that AC-73 could interfere with this interaction to mitigate the consequences of the viral inflammation. However, they do not bring any data to support this hypothesis. At the very least, the determination of CypA in the cardiac tissue on 7 dpi is necessary.   

Minor comments: 

1.       Fig 1: the significance should be pointed at the graph itself and not at the group legend.

2.       Fig 1b:  The AC-73 did not affect the viral load, as it was not significant. To imply that there was an increase, although not significant, is very misleading.

3.       Fig 3: the numbers on the dot plots are too small to read. 

4.       Line 294: CD47 should be CD147

Comments on the Quality of English Language

Minor editing is needed

Author Response

General:

This paper describes how the small molecule inhibitor AC-73, which inhibits CD147 dimerization, can greatly improve the pathological outcome of viral myocarditis. The authors focus on T cell infiltration and activation, and show that compared to mice infected with coxsackievirus B3 (CVB3), AC-73 treated mice T cell infiltration and activation, but not their subset frequencies, are reduced resulting in reduced cardiac inflammation and necrosis.

RE: Thank you for your helpful comments and suggestions on our original manuscript. An item by item response to the comments is included below, those changes are marked in the revised paper. We hope that these revisions address your concerns and requirements.

Major criticism:

1. Introduction: please explain why CD147 dimerization which is disrupted by AC-73, so important to its function, and specifically for its interaction with cyclophilin A (CypA) that is discussed at length in the discussion. 

RE: The dimerization of receptor is known to be an essential mechanism of signaling initiation via receptor tyrosine kinases or receptors associated with cytoplasmic tyrosine kinases [1]. Christian Koch et. al demonstrated that the excess of the mAb, preferentially bind to CD147 monovalently and thus may be unable to dimerize the CD147, which disrupt the function of CD147 [2]. CD147 may require dimerization for initiation of a signaling function. The AC-73 disrupt the dimerization of CD147 affecting the initiation of a signaling function. In addition, as the receptor of CyPA, CD147 binds to CyPA. And we found that the treatment with AC-73 reduced the expression of CyPA. Therefore, we suspect that AC-73 affects the function of CD147, including the CD147-CyPA axis. We have added the words to explain the effect of AC-73 on CD147 function in the Introduction and Discussion of revised manuscript.  

2. The experimental design is not well explained. Why was the endpoint of 7 days post infection chosen? Is the acute-chronic timeline well established in this model?RE: We have established the acute CVB3 model until 14 days post infection. Previous studies took 14 days after infection as the cut-off point of acute and chronic stage of viral myocarditis [3]. And in the CVB3-only model, the peak time point of acute myocarditis on the 7th day after infection [4]. We only focused on the acute inflammation, and at 7 days post infection the immune cells infiltrated significantly in the myocaridium. So we detected the myocardial inflammation and immune response at 7 days post infection. In addition, previous studies showed that the expression of CD147 was up-regulated at 4 days after infection, and reached the maximum expression at 8 days post infection [5]. Therefore, we set the time point of administration on the 4th day after infection to observe the effects of AC-73 on the function and expression of CD147, and further effects on myocardial inflammation. Now we added the words to explain the experimental design in Discussion sections. 

3. CD147 expression was measured only by qPCR at the mRNA level, but its protein levels should also be explored. Moreover, the kinetics of CD147 expression should also be investigated in this model, to make sure that it peaks at 7 dpi.

RE: Thanks for your advice. Peter Seizer et. al have revealed that CD147 showed very low expression in healthy, non-infected mice, and was upregulated in myocardium at 4 days post infection, reaching the maximum expression at 8 days post infection. And immunohistochemical staining confirmed enhanced protein expression of CyPA and CD147 in CVB3-infected myocardium. Although the expression of CyPA and CD147 is both on monocytes and cardiomyocytes. CyPA is predominantly localized at sites of macrophage and T cell infiltration and CD147 was primary detected on myocytes [5]. According to your advice, we detected the CD147 expression by immunofluorescence staining. The results revealed that the expression of CD147 increased in myocardium after CVB3 infection and that treatment with AC-73 had no effect on CD147 expression. For details, please refer to the Results 3.1 (Fig.2). Therefore, we believed that AC-73 affect the function of CD147, which may further reduce the infiltration of immune cells. And the decrease of immune cells also reduced CyPA expression, which reduced the CD147-CyPA axis. 

4. Generally, the authors do not investigate the infiltration of macrophages at all. This is unexplained, and since CD147 is also expressed on macrophages and may be very important for in determining the cardiac microenvironment, this absence is critical.   

RE: Thanks for your advice, we only focused the T cells infiltration before. Indeed, in the acute stage, firstly, the infiltration of natural killer cells and macrophages in myocardium, aggravates myocardial injury. Subsequently, T cells including CD4+ T and CD8+ T infiltrate and induce inflammation or mediate toxicity. Based on your suggestions, we detected the macrophages infiltration in myocardium by immunofluorescence staining and explored the effect of AC-73 in macrophages infiltration. The results revealed that at 7 days post infection, compare to the control group, macrophages increased in myocardium, while the treatment with AC-73 reduced the infiltration of macrophages. Please refer to the Results 3.4 (Fig.8). And previous studies revealed that CyPA is important for macrophage and T cell recruitment [5]. The treatment with AC-73 also decreased the expression of CyPA, which inhibit the recruitment of macrophages induced by CVB3 infection.

5. There is a big difference between checking splenic T cells and determining their state of activation and changes in their subset frequencies, and showing the same data in cardiac infiltrating T cells. Although some effort was made in characterizing cardiac T cells, this was very partial. This reviewer appreciates that the task of isolating T cells from the heart is difficult, but at least it should be supported by some images of immunohistochemical or immunofluorescence staining of CD69 and CD38. 

RE: Thanks for your advice. We detected the CD69 expression in myocardium by immunofluorescence staining. The results showed that the expression of CD69 increased in myocardium after CVB3 infection and the treatment with AC-73 reduced the CD69 expression. The results demonstrated that the treatment with AC-73 decreased the activation of T cells in myocardium. Please refer to the Results 3.2 (Fig.7).

6. The determination of 15 cytokines and chemokines in the plasma is very interesting, but not sufficient. First, the pro-inflammatory cytokines and the chemokines should be explored in situ, that is in the heart tissue lysates.  Secondly, why did the authors specifically choose to investigate only IFNγ, IL-17 and IL-10? Moreover, since cytokines and chemokines are regulated at many different levels, the end-product of proteins should be explored, and not the mRNA.   

RE: Thanks for your advice. We explored the extent of system inflammation by detecting the expression of cytokines and chemokines in the plasma. Then we detected inflammatory cytokines mRNA in myocardium indicating the degree of myocardial inflammation which was consistent with the HE results. We referred the papers which were detected the mRNA expression of cytokines and chemokines in tissues as the inflammatory indicator [6, 7]. In addition, the heart tissue was divided two parts for HE staining and RT-qPCR, now we had not heart tissue lysates for protein level detection. In future, we will focus the protein level of the cytokines in tissue.

Previous studies have revealed that both Th1 and Th17 cells are involved in the pathogenesis of acute viral myocarditis. Acute myocarditis is characterized by a predominantly Th1 and Th17 response [8, 9]. Therefore, we detected the Th1 and Th17 related cytokines in myocardium.

7. The authors speculate a lot about the importance of the binding of CypA to CD147, and hypothesize that AC-73 could interfere with this interaction to mitigate the consequences of the viral inflammation. However, they do not bring any data to support this hypothesis. At the very least, the determination of CypA in the cardiac tissue on 7 dpi is necessary.   

RE: Thanks for your constructive advice, we also detected the CyPA in myocardium by Immunohistochemistry. The results showed that the average density of CyPA in myocardium increased after CVB3 infection. While treatment with AC-73 reduced the inflammatory infiltration, the average density of CyPA also decreased. For details, please refer to the Results 3.1 (Fig.2). In future, we will explore the specific mechanism of CD147-CyPA axis and the effect of AC-73 on CD147-CyPA interaction in CVB3-induced myocarditis.

Minor comments: 

1. Fig 1: the significance should be pointed at the graph itself and not at the group legend.

RE: Thanks for your advice. We have corrected the Figure 1 in revised manuscript.

2. Fig 1b:  The AC-73 did not affect the viral load, as it was not significant. To imply that there was an increase, although not significant, is very misleading.

RE: Thank to your suggestion. We have re-described this part of the result and corrected the words in Discussion section of the revised manuscript.

3. Fig 3: the numbers on the dot plots are too small to read. 

RE: Thanks for your advice. We have relabeled the numbers on the dot plots.

4. Line 294: CD47 should be CD147

RE: We are sorry for our carelessness. We have corrected it.

References

  1. Heldin, C. H., Dimerization of cell surface receptors in signal transduction. Cell 1995, 80, (2), 213-23.
  2. Koch, C.; Staffler, G.; Hüttinger, R.; Hilgert, I.; Prager, E.; Cerný, J.; Steinlein, P.; Majdic, O.; Horejsí, V.; Stockinger, H., T cell activation-associated epitopes of CD147 in regulation of the T cell response, and their definition by antibody affinity and antigen density. International immunology 1999, 11, (5), 777-86.
  3. Esfandiarei, M.; McManus, B. M., Molecular biology and pathogenesis of viral myocarditis. Annual review of pathology 2008, 3, 127-55.
  4. Fairweather, D.; Stafford, K. A.; Sung, Y. K., Update on coxsackievirus B3 myocarditis. Current opinion in rheumatology 2012, 24, (4), 401-7.
  5. Seizer, P.; Klingel, K.; Sauter, M.; Westermann, D.; Ochmann, C.; Schönberger, T.; Schleicher, R.; Stellos, K.; Schmidt, E. M.; Borst, O.; Bigalke, B.; Kandolf, R.; Langer, H.; Gawaz, M.; May, A. E., Cyclophilin A affects inflammation, virus elimination and myocardial fibrosis in coxsackievirus B3-induced myocarditis. Journal of molecular and cellular cardiology 2012, 53, (1), 6-14.
  6. Pan, H.; Zhang, Y.; Luo, Z.; Li, P.; Liu, L.; Wang, C.; Wang, H.; Li, H.; Ma, Y., Autophagy mediates avian influenza H5N1 pseudotyped particle-induced lung inflammation through NF-κB and p38 MAPK signaling pathways. American journal of physiology. Lung cellular and molecular physiology 2014, 306, (2), L183-95.
  7. Zhou, L.; Feng, Z.; Liu, J.; Chen, Y.; Yang, L.; Liu, S.; Li, X.; Gao, R.; Zhu, W.; Wang, D.; Shu, Y., A single N342D substitution in Influenza B Virus NA protein determines viral pathogenicity in mice. Emerging microbes & infections 2020, 9, (1), 1853-1863.
  8. Huber, S. A.; Sartini, D.; Exley, M., Vgamma4(+) T cells promote autoimmune CD8(+) cytolytic T-lymphocyte activation in coxsackievirus B3-induced myocarditis in mice: role for CD4(+) Th1 cells. Journal of virology 2002, 76, (21), 10785-90.
  9. Yuan, J.; Cao, A. L.; Yu, M.; Lin, Q. W.; Yu, X.; Zhang, J. H.; Wang, M.; Guo, H. P.; Liao, Y. H., Th17 cells facilitate the humoral immune response in patients with acute viral myocarditis. Journal of clinical immunology 2010, 30, (2), 226-34.

Reviewer 2 Report

Comments and Suggestions for Authors

In the manuscript 'Inhibitor of CD147 supresses the T cells activation and recruitment in CVB3-induced acute viral myocarditis, submitted by Ruifang Wang and coworkers to Viruses, the authors investigate if AC-73 can alleviate cardiac inflammation in a mouse model. In general, the manuscript is interesting but is at some points unclear.

1.) First, the authors should also indicate in line 28/29 that cardiac inflammation is even present in genetic cases without any virus infection. This was recently shown in the DSC2 transgenic mouse developing severe cardiomyopathy associated with cardiac inflammation (see PMID: 28339476 Brodehl A et al. PLOSone 2017). Please add this information to the introduction.

2.) Please add the culturing conditions for the HeLa cells and indicate the provider of this cell line (Material and Methods Sectiuon).

3.) Please present all data as mean +/- standard deviation (SD) instead of SEM, since the reader is interested in the variation of the data.

4.) Line 96: Please add the manufacturer or provider of this software.

5.) Please add the used antibody concentrations for FACS analysis.

6.) Were the results tested for normal distribution? If not, please use a non-parametric test.

7.) Please increase the size of all figures. Sometimes the writing is so small, that it can not be read.

8.) Please present your data as dot plots instead of bar plots.

9.) Please indicate in Figure 1C if these data were significant.

10.) Figure 2F: The writing is not equally distributed and unsharp. Please correct it. 

11.) Please increase the size of Figure 3. The writing and labelling can not be read.

12.) Please increase the size of Figure 4. The writing and labelling can not be read.

13.) Could you please summarize your findings in a small overview figure?

Good luck with the revision.

Comments on the Quality of English Language

The English is acceptable but should be double checked by a native editor.

Author Response

Response to Reviewer 2

In the manuscript 'Inhibitor of CD147 supresses the T cells activation and recruitment in CVB3-induced acute viral myocarditis, submitted by Ruifang Wang and coworkers to Viruses, the authors investigate if AC-73 can alleviate cardiac inflammation in a mouse model. In general, the manuscript is interesting but is at some points unclear.

RE: Thank you for your helpful comments and suggestions on our original manuscript. An item by item response to the comments is included below, those changes are marked in the revised paper. We hope that these revisions address your concerns and requirements.

1. First, the authors should also indicate in line 28/29 that cardiac inflammation is even present in genetic cases without any virus infection. This was recently shown in the DSC2 transgenic mouse developing severe cardiomyopathy associated with cardiac inflammation (see PMID: 28339476 Brodehl A et al. PLOSone 2017). Please add this information to the introduction.

RE: Thanks for your advice. We have added the words introducing the cause of myocarditis in the Introduction section.

2. Please add the culturing conditions for the HeLa cells and indicate the provider of this cell line (Material and Methods Section).

RE: Thanks for your advice. We have added the related information of HeLa cells in the Materials and Methods Section 2.2.

3. Please present all data as mean +/- standard deviation (SD) instead of SEM, since the reader is interested in the variation of the data.

RE: Thanks for your advice. We have present all data as Mean ± standard deviation (SD) in the revised manuscript.

4. Line 96: Please add the manufacturer or provider of this software.

RE: Thanks for your advice. We have added the manufacturer of this software in the Materials and Methods Section 2.5.

5. Please add the used antibody concentrations for FACS analysis.

RE: Thanks for your advice. We have added the dilutions of antibody for FACS analysis in the Materials and Methods Section 2.8.

6. Were the results tested for normal distribution? If not, please use a non-parametric test.

RE: Thanks for your advice. All data were performed normality tests and homogeneity of variances by Shapiro-Wilk test and Brown-Forsythe test. Then according to the statistical characteristics, we used Kruskal-Wallis test or one-way analysis of variance (ANOVA) to analyze the data. Only the histological score data did not meet the normal distribution. We have corrected the statistical methods for histological score.

7. Please increase the size of all figures. Sometimes the writing is so small, that it can not be read.

RE: Thanks for your advice. We have modified it in the revised manuscript.

8. Please present your data as dot plots instead of bar plots.

RE: Thanks for your advice. We have remade the Figures in the revised manuscript.

9. Please indicate in Figure 1C if these data were significant.

RE: Thanks for your advice. We have remade the Figure 1C in the revised manuscript.

10. Figure 2F: The writing is not equally distributed and unsharp. Please correct it. 

RE: Thanks for your advice. We have remade the Figure 2F in the revised manuscript.

11. Please increase the size of Figure 3. The writing and labelling can not be read.

RE: Thanks for your advice. We have divided Figure 3 into several figures to make the figures clear.

12. Please increase the size of Figure 4. The writing and labelling can not be read.

RE: Thanks for your advice. We have modified the layout of Figure 4 to make it clearer in the revised manuscript.

13. Could you please summarize your findings in a small overview figure?

RE: Thanks for your advice. We have added the graphic abstract. Please see the attachment.

Round 2

Reviewer 1 Report

Comments and Suggestions for Authors

The authors have answered the concerns and substantially revised the manuscript to my satisfaction